# A Novel Fluorescence Aptasensor Based on Magnetic Beads/Gold Nanoparticles/DNA-Stabilized Silver Nanoclusters for Detection of *Salmonella* Typhimurium

**DOI:** 10.3390/foods11040595

**Published:** 2022-02-18

**Authors:** Shiqian Fu, Xinyan Yang, Lidong Pang, Shasha Cheng, Danliangmin Song, Xue Qin, Chaoxin Man, Yujun Jiang

**Affiliations:** 1Department of Food Science, Northeast Agricultural University, Harbin 150038, China; b19100103@neau.edu.cn (S.F.); 15545116996@163.com (X.Y.); a10180384@neau.edu.cn (L.P.); s201001068@neau.edu.cn (S.C.); b20100126@neau.edu.cn (D.S.); b211001006@neau.edu.cn (X.Q.); 2Key Laboratory of Dairy Science, Ministry of Education, Harbin 150030, China

**Keywords:** DNA-stabilized silver nanoclusters, gold nanoparticles, magnetic beads, fluorescence aptasensor, *Salmonella* Typhimurium

## Abstract

*Salmonella* Typhimurium (*S.* Typhimurium) is a globally distributed foodborne pathogen, which can lead to outbreaks of foodborne infectious diseases. It is essential to guarantee food safety by timely and correct detection of *S.* Typhimurium. In this investigation, an original fluorescence aptasensor was constructed to detect *S.* Typhimurium rapidly and sensitively. Through the coupling of magnetic beads, aptamer, and gold nanoparticles (AuNPs), a fluorescence quenching system with a “sandwich structure” was established. The aptamer acted as a link, and its specific binding to *S.* Typhimurium could release AuNPs from the system. Meanwhile, fluorescent DNA-stabilized silver nanoclusters (DNA-AgNCs) were synthesized. The fluorescence intensity changes caused by the fluorescence resonance energy transfer between DNA-AgNCs and AuNPs were utilized to detect *S.* Typhimurium. The purposed aptasensor exhibited high selectivity and sensitivity with a linear response to *S.* Typhimurium, ranging from 3.7 × 10^2^ to 3.7 × 10^5^ cfu/mL. The limit of detection (LOD) was estimated to be 98 cfu/mL within 2 h 10 min. In addition, this method showed excellent application for detection of *S.* Typhimurium in artificially contaminated milk, with LOD reaching 3.4 × 10^2^ cfu/mL. Therefore, the developed fluorescence aptasensor has great potential to identify *S.* Typhimurium in foodstuffs.

## 1. Introduction

*Salmonella* Typhimurium (*S.* Typhimurium), a Gram-negative bacteria pathogen, is associated with outbreaks of foodborne infectious diseases in human and animal hosts worldwide. Its infection leads to human gastrointestinal discomforts, such as nausea, vomiting, diarrhea, and even death in severe cases [1]. The major source of *S.* Typhimurium is the consumption of contaminated eggs, vegetables, fruits, poultry, meat, and milk [2]. The presence of this organism in food poses a severe threat to public safety. Currently, several methods have been proposed to identify *S.* Typhimurium, including traditional culture, molecular detection, and immunological detection. However, each method has its own limitations. The traditional culture method is the most commonly used detection technique, but it is time-consuming and cost-intensive [3]. Several molecular detection methods, such as PCR and LAMP, require expensive instruments and complex DNA extraction procedures [4]. Immunological assays based on antigen–antibody interaction include the structures of antibodies that are deformed under high temperature and batch-to-batch variation in functionality [5]. Hence, it is quite essential to set up a rapid, convenient, and accurate method of detection to ensure food safety, especially for the screening of specific recognition elements of *S.* Typhimurium.

Aptamers are short, single-stranded DNA or RNA molecules. By folding into distinct secondary and tertiary structures, they can combine with strong affinity and selectivity to a broad set of targets, such as small molecules and single cells [6]. As new recognition elements, aptamers offer significant benefits when compared with antibodies used in molecular recognition, including small size, quick synthesis, easy modification, long-term stability, non-toxicity, and low immunogenicity [7]. Some studies have developed aptasensors to test *S.* Typhimurium based on different analytical techniques [8,9,10,11]. Among them, the fluorescent aptasensor is frequently used because of its high sensitivity and accuracy. Most fluorescent molecular beacons, however, such as common organic dyes and quantum dots, have poor photostability, cell toxicity, and large physical size [12]. Thus, a more stabilized and productive fluorescence aptasensor is urgently necessary.

DNA-stabilized silver nanoclusters (DNA-AgNCs), an advanced category of fluorescent nanostructures, have drawn people’s attention extensively on account of their good photostability, excellent biocompatibility, ultrasmall size, high quantum yield, low toxicity, and tunable wavelength. In recent years, DNA-AgNCs, serving as fluorescent probes, have been widely used in biosensors to detect DNA [13], proteins [14], metal ions [15], microRNA [16], small molecules [17], and tumor cells [18]. In addition, gold nanoparticles (AuNPs) are extensively utilized to construct biosensors because of their great biocompatibility, unique features, and unique optical properties [19]. Most importantly, AuNPs have ultrahigh fluorescence quenching capability upon the fluorescent molecular beacons [20]. Some studies about combining the AuNPs and DNA-AgNCs to design fluorescence aptasensors have been reported [21,22]. In these analyses, fluorescence quenching would happen while the fluorophores are linked with AuNPs, resulting in fluorescence resonance energy transfer (FRET) from the organic donor to the AuNPs acceptor.

In this article, an original fluorescence aptasensor was constructed based on magnetic beads/AuNPs/DNA-AgNCs to detect *S.* Typhimurium. Magnetic beads conjugated with complementary DNA (cDNA), aptamer, and AuNPs probe formed a fluorescence quenching system, so DNA-AgNCs can be adsorbed and quenched when added to the system. However, when *S.* Typhimurium was present, its specific recognition with the aptamer led to the release of AuNPs in the fluorescence quenching system, thereby inhibiting the fluorescence quenching of DNA-AgNCs. Therefore, the designed novel fluorescence aptasensor was based on the change in fluorescence intensity to attain a simple, rapid, and sensitive identification of *S.* Typhimurium.

## 2. Materials and Methods

### 2.1. Materials

Sodium tetrahydridoborate (NaBH_4_), HAuCl_4_·3H_2_O, trisodium citrate, and silver nitrate (AgNO_3_) were from Aladdin Reagent Co., Ltd. (Shanghai, China). Streptavidin-modified magnetic beads (SMBs, 10 mg/mL, 1 µm Dynabeads™ MyOne™ T1) were purchased from Invitrogen Biotechnology Co., Ltd. (Carlsbad, CA, USA). All other reagents used in this trial were of analytical grade, which were obtained from Sinopharm Chemical Reagent Co., Ltd. (Shanghai, China). UHT-sterilized milk samples were obtained from a local market (Harbin, China). Oligonucleotide sequences were synthesized by Sangon Biotech Co., Ltd. (Shanghai, China) and purified by HPLC, which are shown in Table 1. Following this, 2 × B&W buffer containing 10 mM Tris-HCl (pH 7.5), 1 mM EDTA, and 2 M NaCl were prepared. Meanwhile, PBS solution (10 mM phosphate buffer saline, pH 7.4), PB solution (20 mM phosphate buffer, pH 7.4), A buffer (10 mM phosphate buffer, 0.1% SDS, pH 7.6), and B buffer (1 mM phosphate buffer, 0.01% SDS, 2 M NaCl, pH 7.6) were also prepared for follow-up study.

### 2.2. Instrumentation

A SpectraMax i3x multifunctional microplate reader from Molecular Devices Co., Ltd. (San Jose, CA, USA) was used to measure the absorption and fluorescence spectra. Transmission electron microscopy (TEM; HT7800, Hitachi, Japan) was applied to record the images of AuNPs. X-ray photoelectron spectroscopy (XPS; ESCALAB 250Xi) from Thermo Scientific (Waltham, MA, USA) and Talos F200X TEM (FEI Company, Hillsboro, OR, USA) were utilized to assess the prepared DNA-AgNCs.

### 2.3. Bacterial Strains and Cultivation Conditions

*S.* Typhimurium ATCC 14028 was selected to conduct the experiment as the target bacteria, and *Staphylococcus aureus* (*S. aureus*) ATCC 25923, *Escherichia coli* O157:H7 (*E. coli* O157:H7) ATCC 25922, *Shigella flexneri* (*S. flexneri*) CMCC 51572, *Cronobacter sakazakii* (*C. sakazakii*) ATCC 29544, *Bacillus cereus* (*B. cereus*) CMCC 63303, *Vibrio parahaemolyticus* (*V. parahaemolyticus*) ATCC 17802, and *Listeria monocytogenes* (*L. monocytogenes*) ATCC 19114 served as non-target strains. All the above bacterial strains were preserved in our lab and were pre-enriched in LB broth and shaken at 37 °C overnight. Then, the cultured bacterial solution was serially tenfold diluted. The appropriate dilutions were coated onto LB agar plates. After cultivation overnight at the same temperature, the colonies were measured to calculate the live count. Three parallel boards were performed for each dilution.

### 2.4. Preparation of SMBs-cDNA1 Complex

The preparation of SMBs-cDNA1 complex was reported in our previous work [23]. First, 1 mg of SMBs was washed three times with 1 × B&W buffer, and redispersed in 200 μL of 2 × B&W buffer. Then, 200 μL of 2 μM cDNA1 was added and reacted at 37 °C with shaking for 1 h. After washing with 1 × B&W buffer, the SMBs-cDNA1 complex was resuspended in PBS solution. Finally, the SMBs-cDNA1 complex with a final concentration of 5 µg/µL was stocked at 4 °C for future trial.

### 2.5. Production of the AuNPs and AuNPs-cDNA2 Probe

The synthesis of AuNPs was conducted according to previous literature [24] with some modifications. Briefly, under vigorous magnetic stirring, 99 mL of ultrapure water was heated to boiling, before adding 1 mL of HAuCl_4_ solution (1%, wt/wt). After heating to boiling, under magnetic stirring, 2.2 mL of sodium citrate solution (1%, wt/wt) was added into the mixture, then boiled and stirred for 15 min until the color turned red. After that, the AuNPs solution was cooled to 25 °C and centrifuged at 12,000× *g* for 20 min at 4 °C. After removal of the supernatant, the prepared AuNPs were dispersed in ultrapure water and stored at 4 °C for further experiments.

The AuNPs-cDNA2 probe was prepared as reported previously [25]. In brief, 8 μL of thiol-modified cDNA2 (100 μM) was mixed with 1 mL of AuNPs solution, followed by incubating at 50 °C for 22 h. Later, 100 μL of A buffer was put into the mixture. After incubation at 50 °C for 60 min, B buffer was added gently for reaction until the NaCl concentration arrived at 0.14 M. The solution was centrifugated at 12,000× *g* for 20 min at 4 °C to remove the excessive cDNA2. Finally, the obtained AuNPs-cDNA2 probe was resuspended in 200 μL of ultrapure water and stored in the dark at 4 °C prior to use.

### 2.6. Synthesis of DNA-AgNCs

The synthesis of DNA-AgNCs was performed on the basis of the method reported in previous literature [26] with little change. First, 89 μL of PB solution containing 1.5 nmol DNA was added to 90 μM AgNO_3_ (6 μL, 1.5 mM), and then incubated in the dark at 4 °C for 20 min. Afterwards, 75 μM NaBH_4_ (5 μL, 1.5 mM) was dispersed into the mixture, followed by vigorous shaking for 1 min. The mixed solution was incubated in the dark at 4 °C for 12 h before use.

### 2.7. Fluorescence Aptasensor Assay of S. Typhimurium

Briefly, the fluorescence quenching system was prepared by mixing 2.5 nM Apt with 30 μg of SMBs-cDNA1 complex and 15 μL of AuNPs-cDNA2 probe in 100 μL of PBS solution, followed by incubation at 37 °C for 90 min. After removal of the supernatant by magnetic separation, the SMBs-Apt-AuNPs were washed with PBS solution, and combined with 200 μL of different concentrations of *S.* Typhimurium, before being incubated at 37 °C for 1 h. The supernatant was discarded and the sediment was resuspended in 80 μL of ultrapure water. Subsequently, 20 μL of DNA-AgNCs was added to the solution. After 10 min, the fluorescence intensity of DNA-AgNCs was recorded by a microplate reader (Excitation at 576 nm; Emission at 643 nm). F0 and F indicated the fluorescence intensity of the absence and presence of *S.* Typhimurium, respectively.

### 2.8. Sensitivity Assessment in Milk Samples

To investigate the practicability of the designed aptasensor in real samples, the detection sensor was applied to detect *S.* Typhimurium in artificially contaminated milk. The UHT sterilized milk samples were tested to confirm the absence of *S.* Typhimurium according to ISO 6579-1: 2017 [27]. The spiked milk samples were arranged by adding different concentrations of *S.* Typhimurium to reach final concentrations ranging from 7.6 × 10^1^ to 7.6 × 10^7^ cfu/mL. Then, the upper whey and fat in the spiked samples were removed by centrifugation, and the pellet was resuspended in the same volume of PBS solution. Finally, the spiked milk samples were analyzed through the experimental process of the proposed fluorescent aptasensor.

## 3. Results and Discussion

### 3.1. Principle of Detection Method

The schematic diagram of the fluorescence aptasensor for testing *S.* Typhimurium is shown in Figure 1. The designed fluorescence quenching system was a sandwich structure including a SMBs-cDNA1 complex, an aptamer, and a AuNPs-cDNA2 probe. The establishment of the system was mainly based on the complementary hybridization between cDNA1 (cDNA2) and the aptamer. In addition, DNA-AgNCs served as the fluorescence signal for this assay. The template used to synthesize DNA-AgNCs consisted of two functional selections. The first selection, containing the cytosine oligonucleotide dC_12_, could bind with Ag^+^ followed by NaBH_4_ reduction to prepare DNA-AgNCs. Some studies have used dC_12_ as an effective basic substance to composite DNA-AgNCs [28,29,30]. The second selection, which included a consecutive adenine (Poly A) and its binding to the AuNPs, quenched the fluorescence of AgNCs using FRET. It has been confirmed that single-stranded DNA could be adsorbed onto the surface of AuNPs readily through van de Waals and hydrophobic interactions [31]. Among the four bases, base A has the highest relative affinity for AuNPs [32].

First, the SMBs-cDNA1 complex, the aptamer, and the AuNPs-cDNA2 probe were incubated to form a fluorescence quenching system. When *Salmonella* was present in the solution, the aptamer specifically coupled with the target goal, leading to the discharge of AuNPs-cDNA2 probe. Therefore, when DNA-AgNCs were added, the complex could fluoresce and not be quenched. However, the SMBs-Apt-AuNPs system inhibited the fluorescence of DNA-AgNCs when *S.* Typhimurium was absent.

### 3.2. Characterization of AuNPs and DNA-AgNCs

The TEM results in Figure 1A indicate that the prepared AuNPs were spherical, well-dispersed, and their size was about 17 nm. The UV−vis absorption spectrum of AuNPs illustrated that it had an absorption peak at approximately 518 nm (Figure 1B). Furthermore, it was observed that the maximum absorption wavelength of the synthesized AuNPs-cDNA2 probe was 526 nm, indicating that it had been successfully prepared.

As revealed from Figure 2A, the prepared AgNCs complex had a characteristic absorption band at 576 nm. When the excitation wavelength was 576 nm, the apparent fluorescence emission appeared at 643 nm (Figure 2B). Meanwhile, the solution displayed an obvious orangey-red color under the UV irradiation. The TEM image indicated that the DNA-AgNCs were nearly global and the mean diameter was 2~5 nm, which is represented by dark clusters (Figure 2C). In addition, the Ag 3d region of XPS spectra showed that the material had two characteristic peaks, which were Ag 3d_5/2_ at 368.0 eV and Ag 3d_3/2_ at 373.8 eV, respectively, indicating the presence of Ag elements (Figure 2D). The above results proved that the DNA-AgNCs synthesis was successful.

### 3.3. Feasibility Testing for Proposed Strategy

As illustrated in Figure 3, fluorescence spectra were utilized to research the feasibility of this detection method. When the aptamer (curve c) or AuNPs-cDNA2 (curve d) was absent, the fluorescence quenching system could not be constructed. Therefore, when the DNA-AgNCs were added, there was no obvious change in the fluorescence intensity. Nevertheless, the fluorescence intensity was significantly reduced in the mixture of SMBs-cDNA1, aptamer, and AuNPs-cDNA2 (curve a). This phenomenon indicates that a complete fluorescence quenching system could lead to effective fluorescence quenching of DNA-AgNCs. Furthermore, a significant fluorescence enhancement was observed when *S.* Typhimurium was present in the sample (curve b). These outcomes demonstrate that the whole system can be applied to detect *S.* Typhimurium.

### 3.4. Optimization of Trial Conditions

In this investigation, the aptamer was simultaneously hybridized with the SMBs-cDNA1 complex and AuNPs-cDNA2 probes in the fluorescence quenching system, and applied to combine with target bacteria. Thus, the concentration of aptamer played an important role in the assay. The lower concentration may result in less sandwich structure (SMBs-cDNA1 complex, aptamer, and AuNPs-cDNA2 probe) formation, and in the absence of *S.* Typhimurium, the fluorescence quenching system cannot be fully functional, resulting in fluorescence of DNA-AgNCs. Whereas, when the system contained a high concentration of aptamer, there was still surplus aptamer after the reaction with *S.* Typhimurium. This resulted in an excess sandwich structure and quenched part of the fluorescence intensity after adding DNA-AgNCs, making it insignificant. From the results in Figure 4A, we found that the fluorescence ratio (1−F0/F) steadily increased with the increasing concentration of aptamer, arriving at its maximum at 2.5 nM, and then decreasing. Based on the above, a 2.5 nM aptamer was used throughout the experiment. Then, the amounts of SMBs-cDNA1 complex and AuNPs-cDNA2 probe were optimized for mixing with the aptamer. From Figure 4B, the (1−F0/F) increased with the growing amount of SMBs-cDNA1, and reached a plateau after 30 μg, before then decreasing at 50 μg. The reason for the decrease may be that the excessive SMBs-cDNA1 influenced the DNA-AgNCs. Figure 4C shows that the (1−F0/F) increased with an increasing amount of AuNPs-cDNA2 probe from 5 to 15 µL, and then remained unchanged at 20 µL. Thus, 30 μg of the SMBs-cDNA1 complex and 15 µL of the AuNPs-cDNA2 probe were selected for subsequent experiments. In addition, to construct the identification for *S.* Typhimurium in a short time, the reaction time of the fluorescence quenching system and the incubation time between the fluorescence quenching system and *S.* Typhimurium were optimized. As shown in Figure 4D,E, the optimal times for both reactions were 1 h.

### 3.5. Sensitivity Detection for S. Typhimurium

The prepared system was conducted to detect different concentrations of *S.* Typhimurium under the best conditions, and the results are shown in Figure 5A. When the *S.* Typhimurium concentration increased from 3.7 × 10^2^ to 3.7 × 10^8^ cfu/mL, the fluorescence intensity gradually increased. Meanwhile, there was a linear relationship between the fluorescence ratio (1−F0/F) and the logarithm of *S.* Typhimurium concentration in the range of 3.7 × 10^2^ to 3.7 × 10^5^ cfu/mL (Figure 5B). The linear regression equation was y = 0.1120x + 0.0307, with a correlation coefficient of R^2^ = 0.9858. According to the calculation formula, the limit of detection (LOD) was 98 cfu/mL, LOD = 3N/S, where N is the standard deviation of the blank sample, and S is the slope of the standard curve. The whole detection could be finished within 2 h 10 min. The reported methods for detecting *S.* Typhimurium with their line ranges and detection time are listed in Table 2. However, some of these approaches are time-consuming or have low detection lines. This method can greatly shorten the detection time on the premise of ensuring the detection limit. Therefore, this fluorescence aptasensor designed in this research provides a simple and rapid novel technique which can achieve low LOD.

To confirm the high selectivity of the developed fluorescence aptasensor, seven common non-*S.* Typhimurium bacteria were selected with the concentration of 10^5^ cfu/mL for detection under the same conditions. As shown in Figure 6, the (1−F0/F) of *S.* Typhimurium was much higher than that of other bacteria, indicating the constructed method had good specificity and selectivity for the target microorganism. This is primarily because of the specific recognition of *S.* Typhimurium by the aptamer.

### 3.6. Sensitivity Test in Artificially Contaminated Milk Samples

To prove the feasibility of the aptasensor, a series of serial dilutions of *S.* Typhimurium were used to prepare artificially contaminated milk. Figure 7A shows that when the *S.* Typhimurium concentration increased from 7.6 × 10^1^ to 7.6 × 10^7^ cfu/mL, the fluorescence intensity increased step by step. A good linear relationship (R^2^ = 0.9754) was realized between the (1−F0/F) and the logarithm of the concentration of *S.* Typhimurium ranging from 7.6 × 10^3^ to 7.6 × 10^6^ cfu/mL (Figure 7B). The LOD was assessed to be 3.4 × 10^2^ cfu/mL. Compared with previously reported methods, this study showed a lower LOD [1,37]. These results illustrate that the fluorescence aptasensor is fit to determinate *S.* Typhimurium in artificially contaminated milk.

To further evaluate the reliability of the aptasensor, the recovery test was carried out by spiking known concentrations of *S.* Typhimurium into milk and the results were listed in Table 3. The recoveries were determined to be 92.9–108.7%, indicating the potential practical application value of the novel fluorescence aptasensor for determination in complex foods.

## 4. Conclusions

In conclusion, a label-free fluorescence aptasensor was set up for rapid, sensitive, and specific detection of *S.* Typhimurium due to the fluorescence change in DNA-AgNCs. The constructed fluorescence quenching system, including magnetic beads, an aptamer, and AuNPs, could effectively quench the DNA-AgNCs. In the presence of *S.* Typhimurium, the system was disrupted, leading to an obvious rise in fluorescence intensity. The results demonstrated that the developed aptasensor showed high selectivity and sensitivity for *S.* Typhimurium detection. At the same time, a good linear relationship was acquired when the concentrations of *S.* Typhimurium ranged from 3.7 × 10^2^ to 3.7 × 10^5^ cfu/mL, with the LOD being 98 cfu/mL. Furthermore, the aptasensor was capable of detecting *S.* Typhimurium successfully in artificially contaminated milk with a LOD of 3.4 × 10^2^ cfu/mL, indicating that this could be regarded as an effective tool for the determination of *S.* Typhimurium in complex foods. Importantly, the established system can detect diverse bacteria by changing the corresponding aptamers.

## Data Availability

Data is contained within the article.

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
