# Peer review of "A Novel Fluorescence Aptasensor Based on Magnetic Beads/Gold Nanoparticles/DNA-Stabilized Silver Nanoclusters for Detection of Salmonella Typhimurium"

_foods, 2022, doi:10.3390/foods11040595_

Round 1

Reviewer 1 Report

The paper entitled "A novel fluorescence aptasensor based on magnetic beads/gold nanoparticles/DNA-stabilized silver nanoclusters for detection of Salmonella Typhimurium" reports the rapid and sensitive detection of S. Typhimurium by constructing a fluorescence quenching system with a sandwich structure through the coupling of magnetic beads, aptamer, and gold nanoparticles (AuNPs). Since it is difficult to detect pathogen at low levels of detection in foods, the results of this study for rapid and sensitive detection of S. Typhimurium is very useful and meaningful in the field of food microbiology and safety.

The research was carried out scientifically according to the objective of the study and the manuscript seems to be relatively well written.

My detail comments are attached below.

  1. The author was written that the detection limit of the developed fluorescent aptamer system was "98 CFU/mL". However, Figure 5 show that the detection standard for fluorescence is 3.7 Í102 CFU/mL. Therefore, it was judged that the detection limit in the manuscript was wrong. Please correct this or write your discussion in the manuscript.
  2. Shigella fexner written in this paper should all be revised to shigella flexneri.
  3. In all figure captions, please unify the use of alphabet letters (uppercase letters or lowercase letters).
  4. Space between measurements and number, and add space between number and °C (revision required throughout the entire manuscript).
  5. Please indicate Figure 3 (a), (b), (c), and(d). "a->d" may make it difficult for readers to understand and reduce the quality of the overall manuscript.
  6. Figure 2(C) is difficult to confirm. Please mark the areas to be emphasized.
  7. Line 230-233: Refine the discussion based on references. Further discussion of the change in fluorescence intensity when the concentration of aptamer is too high or low is necessary.
  8. For discussion of lines 256-258, it is necessary to add detection time to the reported methods (Table 2).

Reviewer 2 Report

Major Comments

The manuscript by Fu et al., describes the use of a complex fluorescence quenching based detection system involving Salmonella specific aptamers and gold nanoparticles. Overall the study is clearly presented but two points require clarification.

1. Is the aptasensor specific for Salmonella Typhimurium only or will it bind other Salmonella species?  

2. As currently written, the advantages of this aptasensor is not clear compared the multitude of novel detection systems which the authors are commended for highlighting in Table 2. The authors suggest the other methods are more time consuming and expensive (L272-3) but provide no evidence that their system is less time consuming or cheaper. Moreover, is it less robust with so many components?

Examples of Minor English corrections:

Do not start sentence with 'And' L14

Replace 'Meanwhile' with 'In addition' L23

Replace 'very emergency' L44

Correct 'binging' L46 and 'determinate' L53

Remove 'to build' L57

Correct 'aroused concern' L60

Remove 'apace' L128 and 'wine' L129

Explain 'aging' L135

Remove 'were' L173 and 'able to' L175

Correct 'almost unconspicuous' L240 and 'added' L241
